# Health care utilization at end of life among patients with lung or pancreatic cancer. Comparison between two Swedish cohorts

**Helena Ullgren**[1,2,3]*, **Per Fransson**[1], **Anna Olofsson**[2], **Ralf Segersvärd**[2,4], **Lena Sharp**[2,5]

**1** Department of Nursing, Umeå University, Umeå, Sweden, **2** Regional Cancer Center, Stockholm, Gotland, Sweden, **3** Theme cancer, Karolinska University Hospital, Stockholm, Sweden, **4** Department of Surgery, CLINTEC, Karolinska Institutet, Stockholm, Sweden, **5** Department of Innovative Care, LIME, Karolinska Institutet, Stockholm, Sweden

* helena.ullgren@umu.se

## Abstract

### Objectives

The purpose was to analyze trends in intensity of care at End-of-life (EOL), in two cohorts of patients with lung or pancreatic cancer.

### Setting

We used population-based registry data on health care utilization to describe proportions and intensity of care at EOL comparing the two cohorts (deceased in the years of 2010 and 2017 respectively) in the region of Stockholm, Sweden.

### Primary and secondary outcomes

Main outcomes were intensity of care during the last 30 days of life; systemic anticancer treatment (SACT), emergency department (ED) visits, length of stay (LOS) > 14 days, intensive care (ICU), death at acute care hospital and lack of referral to specialized palliative care (SPC) at home. The secondary outcomes were outpatient visits, place of death and hospitalizations, as well as radiotherapy and major surgery.

A multivariable logistic regression analysis was used for associations. A moderation variable was added to assess for the effect of SPC at home between the cohorts.

### Results

Intensity of care at EOL increased over time between the cohorts, especially use of SACT, increased with 10%, p<0.001, (n = 102/754 = 14% to n = 236/972 = 24%), ED visits with 7%, p<0.001, (n = 25/754 = 3% to n = 100/972 = 10%) and ICU care, 2%, p = 0.04, (n = 12/754 = 2% to n = 38/972 = 4%). High intensity of care at EOL were more likely among patients with lung cancer. The difference in use of SACT between the years, was moderated by SPC, with an increase of SACT, unstandardized coefficient β; 0.87, SE = 0.27, p = 0.001, as well as the difference between the years in death at acute care hospitals, that decreased (β = 0.69, SE = 0.26, p = 0.007).

**Data Availability Statement:** Data cannot be shared publicly because of rules and regulations in Sweden on register data, data is stored without a personoal identification no in order to protect

autonomy, this according to the ethics approval with no: 2018/2230-31/5.The authors are not allowed to publicly share any of the data. Any requests for data must always go through the Swedish ethics board. They may be contacted here; registrator@etikprovning.se +46 (0)10-475 08 00.

**Funding:** Funding This investigation was supported by grants from the Cancer Research Foundation in Northern Sweden (grant no AMP 18-928). The funding source did not in any way influence design, analysis or interpretation of results.

**Competing interests:** The authors have declared that no competing interests exist.

## Conclusion

These findings underscore an increase of several aspects regarding intensity of care at EOL, and a need for further exploration of the optimal organization of EOL care. Our results indicate fragmentation of care and a need to better organize and coordinate care for vulnerable patients.

## Introduction

Patients with lung and pancreatic cancer are often diagnosed with an already advance disease stage, when the prognosis is poor [1,2] and therefore high-quality palliative and end-of-life (EOL) care are critically important [3]. Previous studies among these patient groups indicate risks for overly intense treatment and care at EOL [4–8] that may impact quality of care. Trends in research during recent decades suggest that intensity of care at EOL is increasing [9–11], which may not always align with patients' values and preferences, for example the wish to die at home [12,13] and to be able prepare and discuss the purpose and priorities at EOL [14,15]. Overly intense EOL care may also create additional burden on health care systems [16] and does not always correspond with improved cancer outcomes, such as extension of life [17,18]. Importantly, early integration of palliative care has shown to enhance the quality of EOL care [17,19,20]. There is no validated tool to measure intensity of care at EOL, but a set of measures is widely used [21,22]. Among these are; aspects of hospitalizations, emergency department (ED) visits, intensive care (ICU) care [16,22,23], late chemotherapy use (14 or 30 days before death) [24], death at acute care hospital [25] and lack of referral to palliative care [7,21].

Recent developments in the treatment of lung and pancreatic cancer have resulted in more favorable treatment outcomes [26,27]. In addition, organizational changes, with the strategic aim to decentralise and shift focus from hospital to primary care, have also affected the regional health care systems in Stockholm, Sweden [28,29]. For instance, there was a rapid expansion of specialized palliative care (SPC) at home, and a decreased number of hospital beds [28,30,31]. Some of these SPC units provide home-based care exclusively, while others also provide in-patient palliative care [30]. In summary, the regional health care system provides limited/no access to palliative care within acute care hospitals. Instead, the SPC at home are organized separately, outside the hospital organizations [30], with a focus on symptom management and support from a multidisciplinary team. Patients with active cancer treatment, who are not referred to SPC at home or in-patient palliative care, often receive symptom management and support from the acute care hospitals, usually by specialist nurses [32], working in the out-patient clinic also responsible for delivering cancer treatments.

To our knowledge, there is no research published exploring trends over time and development of EOL care in the region. With the current study, we aimed to explore trends and predictors of intensity of care at EOL for patients with lung or pancreatic cancer in Stockholm, Sweden. We further explored the differences in intensity of care at EOL for decedents who did or did not receive SPC at home. Our objectives were to identify possible areas of EOL care organization in need for improvement and to open up for discussion on how to best strive for optimal quality of cancer care at EOL.

## Methods

### Study design

We performed a population-based, retrospective study, analyzing registry data among patients with lung or pancreatic cancer, deceased in the years of 2010 and 2017, in the Stockholm region, Sweden. When reporting the data, we followed the STROBE (strengthening the reporting of observational studies in epidemiology) checklist [33]. The Regional Ethical Review board, in Stockholm approved the study (2018/2230-31/5).

**Data sources.** Diagnostic data were retrieved from The Swedish cancer registry [34] and linked with data from the VAL-registry (VårdAnalysLager [CareAnalysisStorage], a regional registry storing data on health care utilization) covering 99% of Stockholm's hospital care (planned and/or unplanned admissions), outpatient care, length-of-stay (LOS) and SPC at home [35].

### Linkage of data

The linking procedure between the two registries was performed using unique personal identification numbers, assigned to all residents in Sweden (by birth or on immigration). This enabled accurate linkage between the registries [36]. After the linking procedure, all data were anonymized in order to ensure confidentiality.

### Study population

We included all identified patients with lung or pancreatic cancer who died in Stockholm, Sweden in the years of 2010 and 2017 respectively. These two time periods were chosen to reflect recent regional health care changes. Exclusion criteria's were; > one cancer diagnosis, no cancer diagnose registered during the last six months in database on health care utilization (assuming to have died of other causes), no outpatient visits, hospitalizations, or visits from SPC at home registered during the last 30 days of life (assumed to have moved outside the region).

**Descriptive variables.** The number of outpatient visits (including visits to receive systemic anticancer treatment [SACT]), radiotherapy (all treatment intentions), receipt of major surgery (excluding percutaneous, diagnostic and endoscopic procedures), and multi-modality treatment (> one treatment modality) during last 30 days, were compared between the cohorts. LOS and hospital admissions were defined as the total number of days and number of times admitted for any type of hospitalization (acute-, geriatric-, palliative or long-term care, rehabilitation).

**Variables to measure intensity of care during the last 30 days of life.** We adapted and used a well-established framework for measuring intensity of EOL care [21]. For the last 30 days of life, we determined whether the patient had received SACT, visited the ED, ICU care, hospitalized > 14 days, referred to SPC at home or died at an acute care setting. We used a summary score adapted from previous work, to measure the intensity of EOL care [9,37]; giving each positive measure of intensity of EOL care one point, thus a maximum score of six. The variable 'number of hospitalizations', usually included in the framework, was excluded as only two patients out of 1450 admitted to hospital had > one hospitalization the last 30 days of life.

**Explanatory variables.** Our explanatory variables of interest included age, gender, patient deceased (in 2010 or 2017), pancreatic or lung cancer, and receipt of SPC at home.

**Statistical analysis.** We compared differences in patient characteristics between the groups (deceased in 2010 or 2017) with Wilcoxon two-sample test for continuous variables,

and Chi-squared test for categorical variables. We performed multivariable logistic regression analysis for each outcome of intensity of EOL care, and a multinomial logistic regression on the summary score of intensity of care, categorized as 0+1 (low score), 2+3, and 4+5 (high score), adjusting for age, gender, diagnosis, and year. To describe survival in this cohort, Kaplan-Meier were applied to illustrate the comparisons between years, diagnosis and SPC at home (Y/N). Further, we performed a log-rank test to compare the difference between the survival curves.

We assessed if SPC at home moderated the difference in intensity of EOL care between the cohorts (deceased in 2010 or 2017). Firstly, we performed separate logistic regression models for each outcome; (SACT, ED visits, ICU care, death at acute care hospital and LOS > 14 days), with SPC at home as a moderating variable. We adjusted for age and gender in all models. Significant interactions were followed by further subgroup analysis for those outcomes by dividing the groups by cancer diagnosis (lung or pancreatic cancer). A significance level of <0.05 was used for all statistical analysis and all statistical tests were two sided. Analyses were performed using statistical software R (version 3.6.2) and IBM SPSS statistics version 24.

## Results

In total, 1726 patients were included (lung cancer, n = 1238, 72%; pancreas cancer, n = 488, 28%) in the final sample. The median age was 72 years, (range 37–98), and relatively equally distributed between men and women (Table 1). Between 2010 and 2017, we found a difference between the diagnose groups; pancreatic cancer increased from (n = 189/754 = 25% to 299/972 = 31%) while lung cancer decreased (n = 564/754 = 75% and 673/972 = 69%, p = 0.009). The median survival time of lung cancer was significantly higher in 2017, compared with 2010 (6.8 and 9.2 month respectively, p<0.001). We observed no significant difference for patients with pancreatic cancer in median survival. In contrast to 2017, where we did not find any survival benefit in the group with SPC at home, we found an improved survival in the 2010 cohort with SPC at home (7.6 vs 4.5 months; p<0.001).

### Overall health care utilization comparison between years

Health care utilization in general varied between 2010 and 2017 (Table 1). While out-patient visits increased (from 56 to 79%; p <0.001) whereas median LOS decreased (from 16 to 14 days; p = 0.012) as well as the use of radiotherapy decreased (from 13 to 7%; p<0.001). We found no significant differences for major surgery, multimodality treatment, hospitalizations or death at home. However, death in non-acute care settings (geriatric, palliative or elderly care outside acute care hospital) increased from 49% to 56% (p = 0.007).

### Overall trends in intensity of care at End-of-life between years

Whereas deaths in acute care settings decreased from 2010 to 2017 (29 to 20%, p<0.001), the proportion of patients receiving SACT increased (14 to 24%; p<0.001) as well as both ED visits (3 to 13%, p<0.001) and ICU care (2 to 4%, p = 0.007). As shown in Fig 1, SPC referral increased (39 to 51%, p<0.001). No difference was found in the summary score of intensity of care, between the years (score; median 2, min-max [0–4] and 2 [0–5], respectively). By conducting a multinomial logistic regression adjusted for age, gender and diagnose, we found younger age to be associated with higher score of intensity of care (OR 1.07 [1.04–1.09]; p<0.001). Furthermore, when performing multivariable logistic regression analysis, patients with pancreatic cancer were less likely to die in acute care hospital (OR O.54 [0.41–0.71], p<0.001), visit the ED (OR 0.61 [0.39.0.94], p = 0.031 and receive ICU care (OR 0.44 [0.19–0.90], p = 0.035, but more likely to receive SPC at home (OR 1.54 [1.25–1.91], p<0.001.

**Table 1. Description of the total sample and difference in End-of-life care between the cohorts in the years of 2010 and 2017.**

| Year | 2010 | 2017 | Difference between years +/- in % | P-value* | Total cohort |
|---|---|---|---|---|---|
| **Number of patients, n (%)** | 754 (44) | 972 (56) | | | 1726 (100) |
| **Median age [min-max]** | 70 [38–96] | 73 [37–98] | NA | <0.001 | 72 [37–98] |
| | **N (%)** | **N (%)** | | | **N (%)** |
| **Female** | 368 (49) | 482 (50) | +1% | 0.747 | 850 (49) |
| **Male** | 386 (51) | 490 (50) | -1% | 0.747 | 876 (51) |
| **Pancreatic cancer** | 189 (25) | 299 (31) | +6% | 0.009 | 488 (28) |
| **Lung cancer** | 564 (75) | 673 (69) | -6% | 0.009 | 1237 (72) |
| **Systematic anticancer treatment (SACT)** | 102 (14) | 236 (24) | +10% | <0.001 | 338 (20) |
| **Radiotherapy** | 96 (13) | 66 (7) | -6% | <0.001 | 162 (9) |
| **Major surgery** | 17 (2) | 14 (1) | -1% | 0.206 | 31 (2) |
| **Multimodal treatment** | 20 (3) | 26 (3) | 0 | 0.151 | 46 (3) |
| **Emergency dep visit** | 25 (3) | 100 (10) | +7% | <0.001 | 125 (7) |
| **ICU/intensive care** | 12 (2) | 38 (4) | +2% | 0.004 | 50 (3) |
| **Hospital admission total** | 644 (85) | 806 (83) | -2% | 0.162 | 1450 (84) |
| **Unplanned hospital admission**** | 481 (64) | 614 (63) | -1% | 0.801 | 1095 (63) |
| **Median days length-of-stay in hospital** | 16 [0–31] | 14 [0–31] | -2% | 0.012*** | 15 [0–31] |
| **Length-of-stay > 14 days** | 391 (52) | 473 (49) | -3% | 0.190 | 864 (50) |
| **Out-patient visits** | 423 (56) | 769 (79) | +23% | <0.001 | 1192 (69) |
| **Death at acute care hospital** | 216 (30) | 197 (21) | -9% | <0.001 | 413 (24) |
| **Death at hospital***** | 371 (49) | 542 (56) | +7% | 0.007 | 913 (53) |
| **Death at home** | 124 (16) | 183 (19) | +3% | 0.223 | 307 (18) |
| **Missing place of death** | 43 (6) | 50 (5) | -1% | 0.687 | 93 (5) |
| **Palliative care** | 294 (39) | 497 (51) | +12% | <0.001 | 791 (46) |
| **Median score intensity of care [min-max]***** | 2 [0–4] | 2 [0–5] | 0 | | 2 [0–5] |

*Pearson chi-square.

**Percentage of the patients with hospital admissions.

*** Mann-Whitney-Wilcoxon test.

****Any hospitalization apart from acute care setting.

*****Score of six measures for intensity of care at End-of-life; iv chemo, emergency room visit, length of stay more than 14 days, ICU/intensive care, non-referral to palliative care or death at acute care hospital → this is all for the last 30 days of life.

## Outcomes by specialized palliative care at home

In a multivariable logistic regression model, when adding SPC at home as a moderation variable (SPC at home yes/no x year), we found a larger increase in SACT between the years for patients with SPC; unstandardized coefficient (β; 0.87, SE = 0.27, p = 0.001 (Fig 2). Further, in a sub-group analysis of the diagnostic groups, we found no moderation effect in the group with SPC at home on SACT use, among patients with pancreatic cancer. However, we found a moderation effect in the group of patients with lung cancer (β = 0.94, SE = 0.32, p = 0.003), again a larger increase of SACT in the group with SPC at home.

We also found that SPC at home moderated the difference between years, regarding death in acute care hospital (β = 0.69, SE = 0.26, p = 0.007), Fig 3. Specifically, the difference in death in acute care hospital, between the years of 2010 and 2017 decreased in the group with SPC at home. In the subgroup analysis, we found no statistically significant moderation effect by SPC at home in the group with pancreatic cancer. However, among patients with lung cancer, there was a moderation effect; (β = 0.80, SE = 0.30, p = 0.007), with a decreased difference between the years if SPC at home, regarding death at acute care hospital.

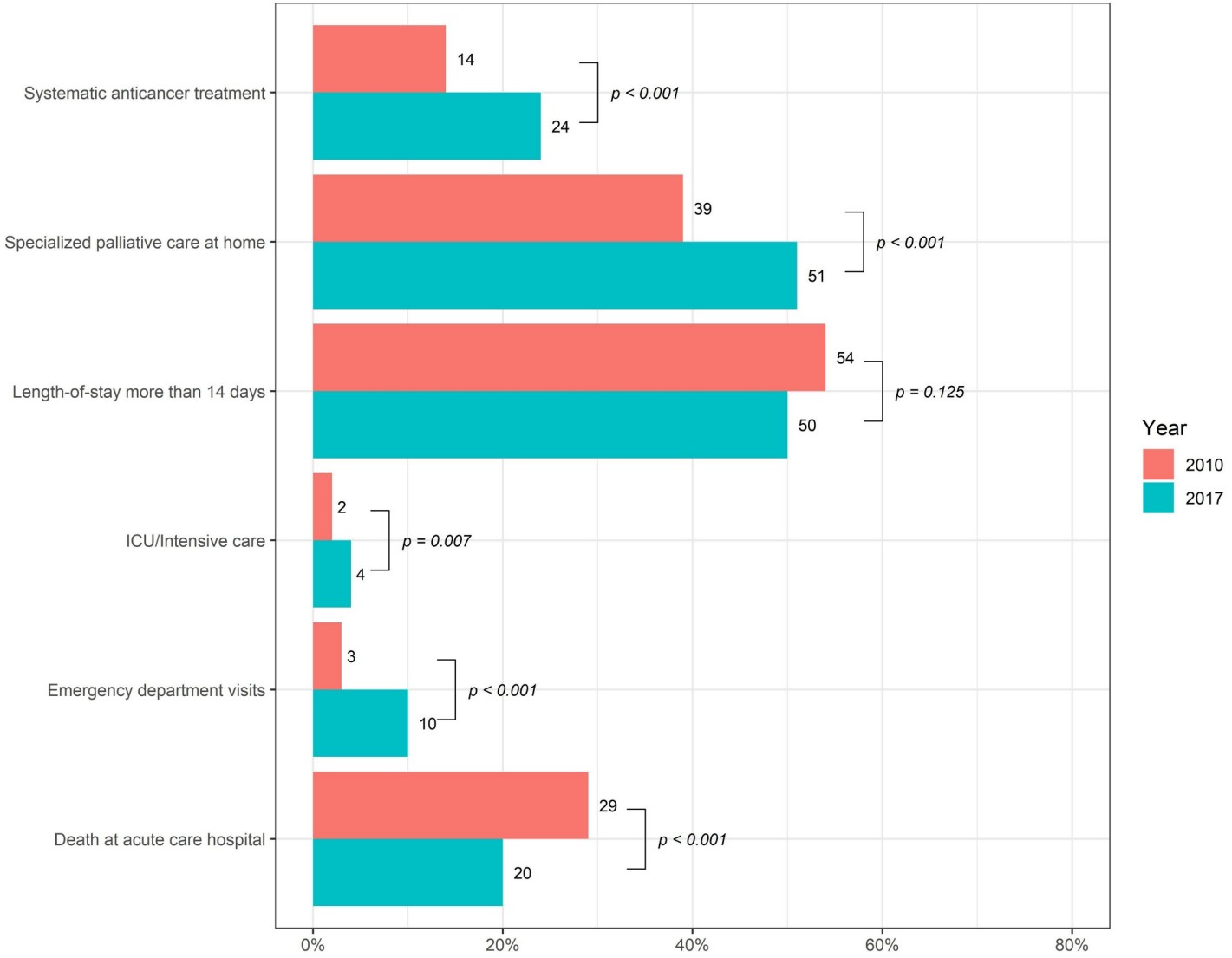

**Fig 1. Trends in intensity of End-of-life care between the years 2010 and 2017.**

There was no statistically significant moderation effect by SPC at home regarding LOS, visits at the ED nor for ICU care.

## Discussion

In this large, population-based register study, we found that several aspects of intensity of care at EOL increased over time for patients with lung- and pancreatic cancer in the Stockholm region, Sweden. In fact, most aspects of health care utilization (SACT, ED visits, ICU care, outpatient visits, death at hospital and referral to SPC at home) increased. However, some aspects decreased (radiotherapy, median hospital LOS, and death at acute care hospital), pointing towards a shift, from inpatient to outpatient care, as intended by regional stakeholders [28,29]. These results, confirmed with the summary score of the intensity of care being stable between the years, reflecting the mixed trends with both increase and decrease of intensity of care measures.

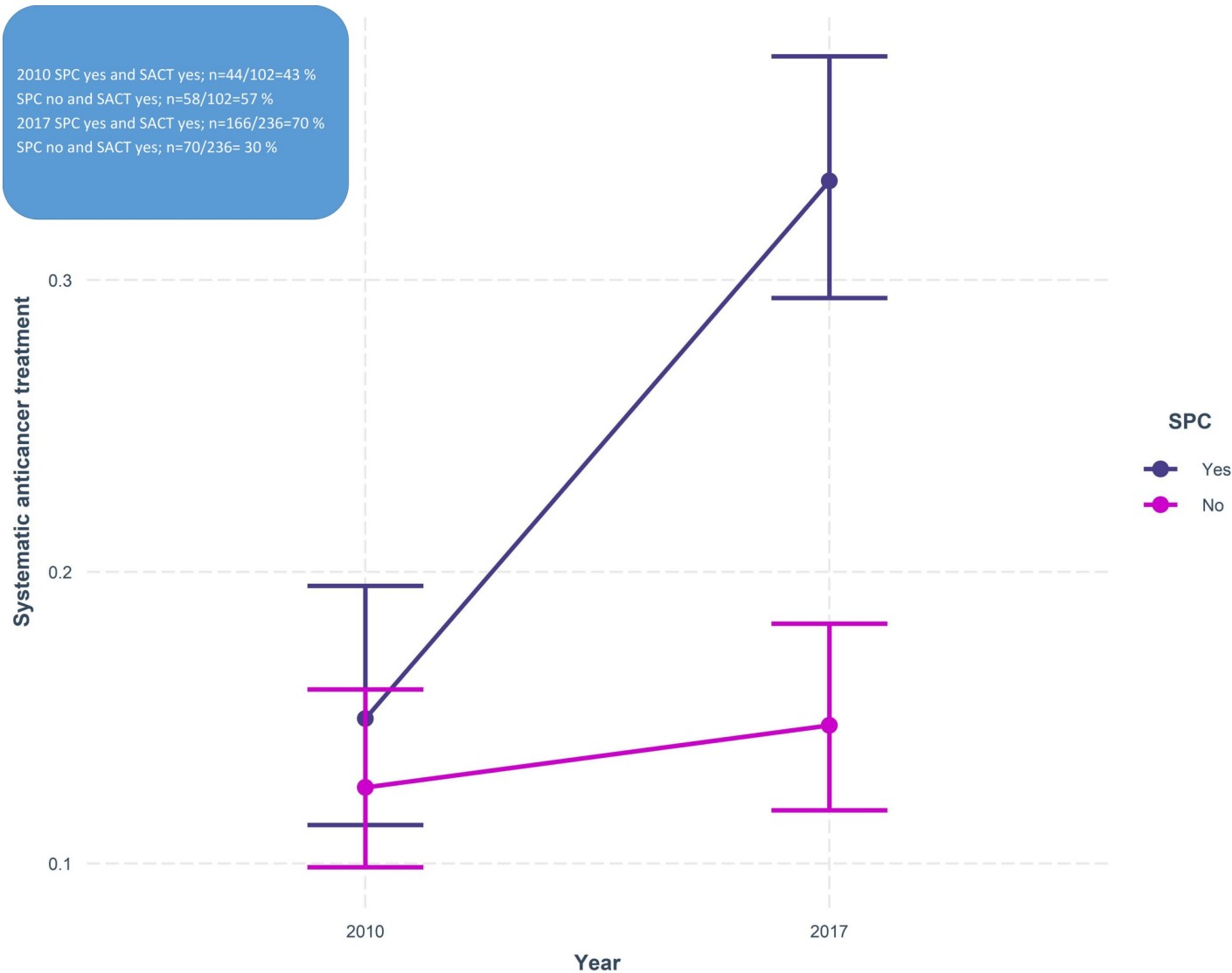

2010 SPC yes and SACT yes; n=44/102=43 %
SPC no and SACT yes; n=58/102=57 %
2017 SPC yes and SACT yes; n=166/236=70 %
SPC no and SACT yes; n=70/236= 30 %

**Fig 2. Moderation effects of specialized palliative care (SPC) at home on the difference between the years of 2010 and 2017 in patients receiving systematic anticancer treatment last 30 days of life.**

An interesting finding was that patients with pancreatic cancer care were associated with lower intensity of care (ICU care, ED visits, death at acute care hospital) compared with patients with lung cancer. Additionally, patients with pancreatic cancer were more likely to receive SPC at home, indicating a less fragmented care trajectory. Possible explanations may include different treatment protocols for the two patient groups. However, we found no significant differences in proportion of SACT use at EOL between the groups. Another explanation to differences between the groups, could be disease-specific symptoms, such as respiratory problems, most likely more common among patients with lung cancer [6], which might challenge SPC teams to provide high-quality care in home settings. Previous research [4] have indicated that patients with lung cancer are particularly at risk for high levels of intensity of care at EOL.

The increased use of SACT in our study may have several explanations, possibly related to more treatment options [26]. Chemotherapy or SACT is the most frequently used measure of quality of EOL care [4,22]. Our results align with recent studies in Australia [24] and France

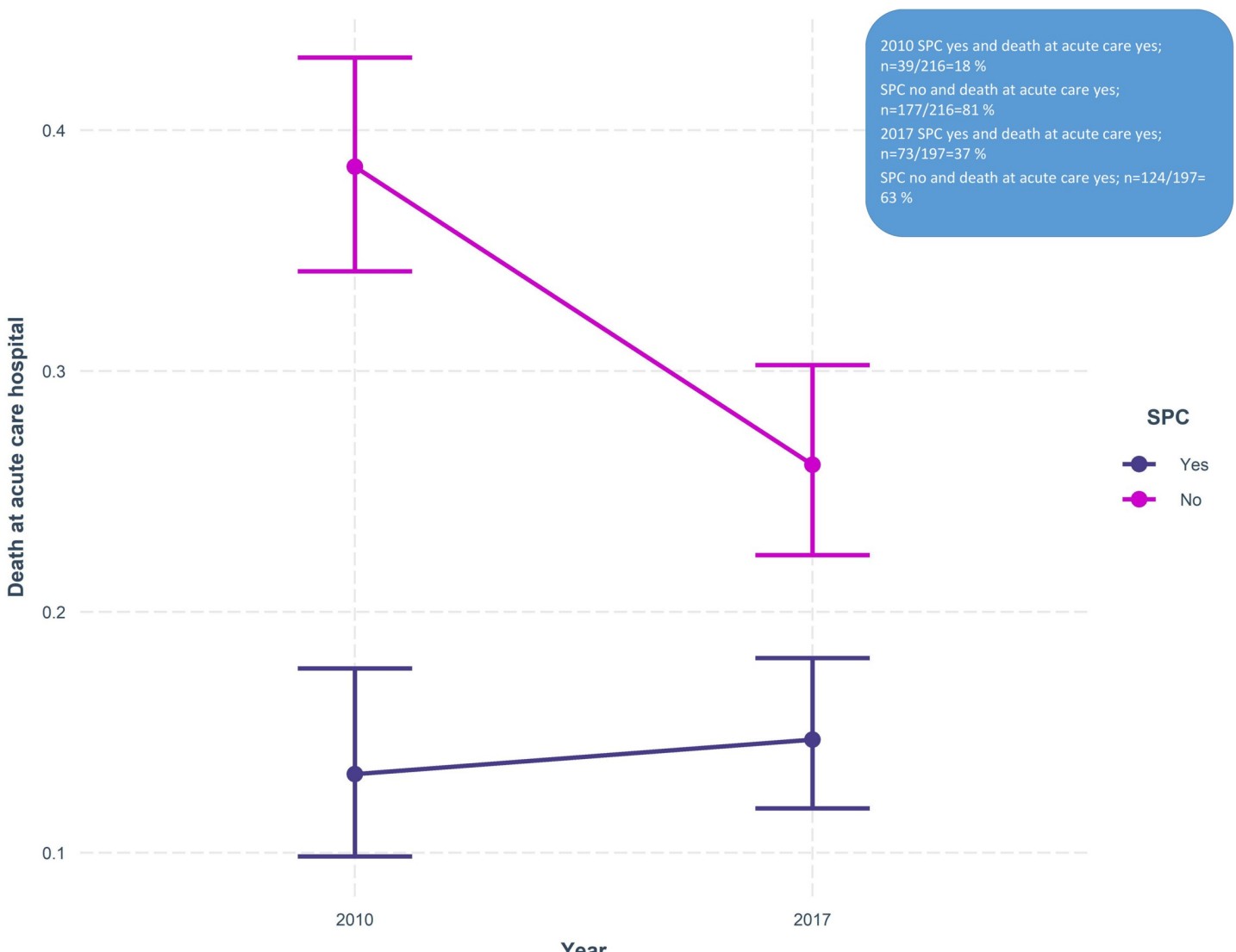

2010 SPC yes and death at acute care yes; n=39/216=18 %
SPC no and death at acute care yes; n=177/216=81 %
2017 SPC yes and death at acute care yes; n=73/197=37 %
SPC no and death at acute care yes; n=124/197= 63 %

**Fig 3. Moderation effects of specialized palliative care (SPC) at home on the difference between the years of 2010 and 2017 in patients dying in acute care hospital.**

[23], both showing similar levels of SACT use, but also local variations. Nguyen et al., 2020 [24] found an increase in use of immunotherapy, but a decrease in other forms of SACTs, among a variety of cancer patients at EOL. The use of potent and highly toxic cancer treatments during the last month of life can be unnecessary and even unethical, but the purpose of these treatments may be to reduce symptoms, and therefore appropriate. Further consequences of overusing SACT may be less focus on important discussions and conversations regarding EOL, such as preferred place of death [38] and goals of care. Previous research shows that most patients want to prepare and focus on quality of life near EOL [15]. In addition, research indicates that unclear decisions on the intent of cancer treatment may led to an overuse near EOL [38]. Even if increased use of SACT at EOL may reflect patients' preferences for life-prolonging treatments, particularly in case of recent diagnosis, as indicated by Voogt et al., [14], our results stresses the importance of integrating a palliative care approach, including conversations on goals of care and preferred place of death.

One explanation of the increased ED and ICU care may also be related to more extensive use of SACT at EOL [39]. We didn't collect data on reasons for ICU/ED care but previous research has indicated similar results for the same patient groups [40–42]. Symptoms such as pain and dyspnea, complications from treatment and comorbidities [43,44] seems to be important drivers for ED visits at EOL [40]. Kaufman et al., [45] concluded in a study among patients with cancer that better psychosocial support and coordination of care are important factors to avoid acute care at EOL.

Another interesting shift was that a larger proportion of patients died in non-acute care settings (geriatric, palliative or elderly care, outside the acute care hospital) in the 2017 cohort. Consequently, death at home did not increase over time, which in previous research have been reported as the preferred place of death [12,13]. We can only speculate on why, as a much larger proportion of patients received home-based SPC in the later cohort. In contrast, previous research indicate that palliative care interventions increase the likelihood of dying at home [46]. Dying in hospital had a negative impact of QoL at EOL in a recent national Swedish study [25], where 25% died in hospital. However, this study included all inpatient settings, and therefore not entirely comparable to our findings. In addition, death in acute hospital may increase the risk mental health issues among family caregivers [47]. So, even if it is encouraging that death at acute care hospital decreased over time in our study, the shift is to other care settings, rather than to home.

In terms of organization of care at EOL, our results show increased use of SPC at home over time. Despite this, our findings raise questions on how well integrated care is, since the use of SACT, ED visits, ICU care and death in other care settings increased. The results therefore indicate that SPC at home does not contribute to the integration of care, which is an important care quality factor [48]. Specifically, the increased measures of intensity of care at EOL means that the patients interact in parallel with both acute and palliative care teams, risking a fragmented care trajectory and unnecessary care transitions at EOL [49]. Considering that every care transition (including transfer of responsibility of care, not only a physical transition), may impact patient safety, this is not without consequences [50]. Further, one way of measuring both the performance of the health care system and the level of integration between them are different aspects of unplanned care [51]. In addition, the results from a recent survey in the region indicates poor integration between acute cancer care and palliative care [52]. Even if the reasons for more frequent referral to SPC might be to strengthen support and symptom management during cancer treatment. This may be related to the separated health care systems for acute and palliative care in the region. The SPC at home in the region, is defined as specialized palliative care, including team members with special training and competence in the field, in contrast to the more general palliative care teams [53]. However, the rapid expansion of SPC during the years of 2010 to 2017, may have challenged the availability of training, and thereby competence levels in the SPC teams. We can only speculate, but to increase availability and allowing early referral to SPC for all stages of cancer, may have influenced the results, as the increasing number of patients during cancer treatment in SPC, and also a more mixed patient load. In a previous study [54], we concluded that the communication between the acute and palliative health care organizations were inadequate and impacting unplanned acute hospital admissions. The separated health care systems, with no formal integration, might also complicate communication between health care providers and patients, regarding levels and goals of care, creating uncertainty that results in unnecessary ED visits/ICU care. In a systematic review, the authors conclude that there is a knowledge gap regarding the most favorable model of proving palliative care and that evaluations tend to lack in description and quality [55].

When comparing survival rates between the two cohorts, we found improved survival for patients with lung but not pancreatic cancer, as well as improved survival in the group with SPC at home (in 2010). Previous research has also found improved survival among lung cancer patients receiving palliative care [17]. We cannot draw any specific conclusions on the reasons behind, as we lack data on cancer stage at point of diagnose. This may be a result of improved treatments or selection of patients for SPC, but other factors may also influence the results.

The main limitations in this study are related to the fact that we are lacking sociodemographic data, as well as data on co-morbidities and symptom burden, which might have influenced the results. The lack of data on disease stage at diagnosis makes it impossible to draw relevant conclusions on differences related to survival. However, this was not the purpose of the study.

In addition, there is always risks in large cohort studies for statistically significant results without practical relevance. However, the population-based design and excellent coverage of the registry data is strengthening our results, providing rich and important descriptions of EOL care for two relatively large patient groups.

## Conclusions

Collectively, these findings underscore an increase in most aspects of health care utilization at EOL in more recent years, and the need for further exploration of the optimal organization of EOL care. Our findings also provide important insights, since several of the changes in health care impacting EOL care are not unique for this region and might be applicable elsewhere [56]. Even if fewer patients in the latter cohort died in acute care settings, they also received more cancer treatment, ICU care and visited the ED more frequently, indicating high intensity of care and higher health care utilization. These factors may have a negative impact on quality of care as well as contribute to fragmentation of EOL care. We hope that our results help health care organizations and stakeholders to better organize and coordinate care for vulnerable patients, e g lung cancer as well as more focus on distressing symptoms and psychosocial issues.

## Supporting information

**S1 Fig. Description of the selection process for the included patients.** A Diagram displaying the inclusion process and reasons for exclusion, as well as the total cohort included. (DOCX)

## Author Contributions

**Conceptualization:** Helena Ullgren, Per Fransson, Anna Olofsson, Ralf Segersvärd, Lena Sharp.

**Data curation:** Helena Ullgren, Per Fransson, Anna Olofsson, Lena Sharp.

**Formal analysis:** Helena Ullgren, Per Fransson, Anna Olofsson, Ralf Segersvärd, Lena Sharp.

**Funding acquisition:** Helena Ullgren.

**Investigation:** Helena Ullgren, Per Fransson, Ralf Segersvärd, Lena Sharp.

**Methodology:** Helena Ullgren, Per Fransson, Anna Olofsson, Ralf Segersvärd, Lena Sharp.

**Project administration:** Helena Ullgren, Lena Sharp.

**Resources:** Helena Ullgren, Per Fransson, Anna Olofsson, Ralf Segersvärd, Lena Sharp.

**Software:** Helena Ullgren, Per Fransson, Anna Olofsson, Lena Sharp.

**Supervision:** Per Fransson, Ralf Segersvärd, Lena Sharp.

**Validation:** Per Fransson, Anna Olofsson, Ralf Segersvärd, Lena Sharp.

**Visualization:** Helena Ullgren, Anna Olofsson, Lena Sharp.

**Writing – original draft:** Helena Ullgren, Per Fransson, Anna Olofsson, Ralf Segersvärd, Lena Sharp.

**Writing – review & editing:** Helena Ullgren, Per Fransson, Anna Olofsson, Ralf Segersvärd, Lena Sharp.

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
