## [Decision Letter · Decision Letter 0]

19 May 2021

PONE-D-21-08049

Health care utilization at end of life among patients with lung or pancreatic cancer. Comparison between two Swedish cohorts

PLOS ONE

Dear Dr. Ullgren,

Thank you for submitting your manuscript to PLOS ONE. After careful consideration, we feel that it has merit but does not fully meet PLOS ONE’s publication criteria as it currently stands. Therefore, we invite you to submit a revised version of the manuscript that addresses the points raised during the review process:

Specifically:

1-For readers who are not familiar with the Swedish health care system, please elaborate on the quality and accessibility of the study population to the specialized palliative care (SPC)  as in/out- patient. What would SPC at the home cover, which is different from in-patient palliative care services and resulting in decreased numbers of acute hospital beds?

2- Please be specific on the integration of EOL care and how the increased use of SACT, ED visits, ICU care, and death in other care settings endanger this integration?

3- Please address comments by reviewers #1,2

We look forward to receiving your revised manuscript.

Kind regards,

Amir Radfar, MD,MPH,MSc,DHSc

Academic Editor

PLOS ONE

Journal Requirements:

2) Please provide additional details regarding participant consent. In the ethics statement in the Methods and online submission information, please ensure that you have specified (1) whether consent was informed and (2) what type you obtained (for instance, written or verbal, and if verbal, how it was documented and witnessed). If your study included minors, state whether you obtained consent from parents or guardians. If the need for consent was waived by the ethics committee, please include this information.

3)  We note that you have indicated that data from this study are available upon request. PLOS only allows data to be available upon request if there are legal or ethical restrictions on sharing data publicly. For information on unacceptable data access restrictions, please see http://journals.plos.org/plosone/s/data-availability#loc-unacceptable-data-access-restrictions.

4) We note that you have included the phrase “data not shown” in your manuscript. Unfortunately, this does not meet our data sharing requirements. PLOS does not permit references to inaccessible data. We require that authors provide all relevant data within the paper, Supporting Information files, or in an acceptable, public repository. Please add a citation to support this phrase or upload the data that corresponds with these findings to a stable repository (such as Figshare or Dryad) and provide and URLs, DOIs, or accession numbers that may be used to access these data. Or, if the data are not a core part of the research being presented in your study, we ask that you remove the phrase that refers to these data.

Reviewers' comments:

Reviewer's Responses to Questions

**Comments to the Author**

1. Is the manuscript technically sound, and do the data support the conclusions?

Reviewer #1: Yes

Reviewer #2: Yes

2. Has the statistical analysis been performed appropriately and rigorously? 

Reviewer #1: Yes

Reviewer #2: Yes

3. Have the authors made all data underlying the findings in their manuscript fully available?

Reviewer #1: Yes

Reviewer #2: Yes

4. Is the manuscript presented in an intelligible fashion and written in standard English?

Reviewer #1: Yes

Reviewer #2: Yes

5. Review Comments to the Author

Reviewer #1: The study emphasizes the importance of integrating a palliative care approach, including conversations about care goals and preferred place of death. In addition, better psychosocial support and care coordination are important factors in avoiding acute care at EOL. Another interesting change was the fact that a higher percentage of patients died in non-acute settings. In terms of the organization of EOL care, the results show an increase in the use of SPC at home over time. Despite this, our findings raise questions about how well-integrated care is, as the use of SACTs, ED visits, ICU care and death in other care settings has increased. the study found that communication between acute and palliative health organizations is inadequate and has an impact on unplanned acute hospital admissions.

Research which I hope will be a stimulus for health care organizations and stakeholders to better organize and coordinate care for vulnerable patients

Reviewer #2: An extremely clear and well written report.

I do wonder about the availability of data regarding provision of care and potential differences between 2010 and 2017: E.g. in the discussion section, page 11 starting in line 244: could authors comment regarding provision of care, mainly a) healthcare professionals palliative care training differences between 2010 and 2017 and b) the existing number of home care teams between 2010 and 2017 across the region and if these could have affected results in any way

There are quite a few typos. This is by no means an exhaustive list.

Page 3 line 53 “and to be to able prepare and discuss the purpose and priorities at EOL”

Page 3 line 57 “but a set of measures is widely used”

Page 4 line 86 “Diagnostic data were retrieved”

Page 4 line 92 “patients with lung or pancreatic cancer who died in Stockholm”

And so on

6. PLOS authors have the option to publish the peer review history of their article (what does this mean?). If published, this will include your full peer review and any attached files.

Reviewer #1: **Yes: **Cascioli Marta

Reviewer #2: No

---

## [Author Response · Author response to Decision Letter 0]

8 Jun 2021

PONE-D-21-08049

Health care utilization at end of life among patients with lung or pancreatic cancer. Comparison between two Swedish cohorts

PLOS ONE

Dear Dr. Ullgren,

Thank you for submitting your manuscript to PLOS ONE. After careful consideration, we feel that it has merit but does not fully meet PLOS ONE’s publication criteria as it currently stands. Therefore, we invite you to submit a revised version of the manuscript that addresses the points raised during the review process:

Specifically:

1-For readers who are not familiar with the Swedish health care system, please elaborate on the quality and accessibility of the study population to the specialized palliative care (SPC) as in/out- patient. What would SPC at the home cover, which is different from in-patient palliative care services and resulting in decreased numbers of acute hospital beds?

Thank you for pointing this out. This was a strategic decision from health care officials, to decentralise cancer away from the hospitals to get closer to the patients. As a result, the number of hospital beds were reduced. SPC at home should include inpatient services, at hospice for instance. We have tried to clarify the introduction in the second paragraph, page 4, starting from line 66 throughout 75 marked in red. 

2- Please be specific on the integration of EOL care and how the increased use of SACT, ED visits, ICU care, and death in other care settings endanger this integration?

Thank you, this is important and we have tried to clarify and elaborated on this in the discussion section, page 12, paragraph 4, starting on line 264 to 271- marked in red. 

3- Please address comments by reviewers #1,2

We have addressed these, see below, under each reviewers comments. 

We also took the liberty to (just after Table 1) add footnotes that was not visible in manuscript submitted, see marked in red at line 161-166, as well as clarification of a sentence in the abstract, that was not clear, page 3, first paragraph, line 41-43

Thank you! We have read those and re-named the files accordingly, and added in the end of the abstract, page 20, “Supporting information” on S1Fig. line 512-514.

2) Please provide additional details regarding participant consent. In the ethics statement in the Methods and online submission information, please ensure that you have specified (1) whether consent was informed and (2) what type you obtained (for instance, written or verbal, and if verbal, how it was documented and witnessed). If your study included minors, state whether you obtained consent from parents or guardians. If the need for consent was waived by the ethics committee, please include this information.

Thank you for noticing that we were not clear. We have no informed consents, since we only used register data on deceased patients. 

The ethics committee does not require informed consents for using registry data, and all data was anonymized before analysis. We have in the manuscript, in methods, page 5, third paragraph, on line, 97-100 added a small section, explaining the linking procedure, and how we did to ensure autonomy. We have not accessed patients’ medical records- only data pulled from registries. To be more clear, we have changed a word in methods section, page 5, second paragraph, line 93- from database to registry (marked in red).

3) We note that you have indicated that data from this study are available upon request. PLOS only allows data to be available upon request if there are legal or ethical restrictions on sharing data publicly. For information on unacceptable data access restrictions, please see http://journals.plos.org/plosone/s/data-availability#loc-unacceptable-data-access-restrictions.

We are sorry and apologize, for being unclear. We do believe that all relevant data are within the manuscript and supporting files. It isn’t allowed under Swedish Ethical law to share data, when it involves information on humans, however we have provided the contact information to the Ethics board, since all requests for data must go through them. 

I will address this in the cover letter, and as mentioned above, no data may be shared according to Swedish ethical regulations. Any request must go through the Ethics board. They may be contacted here; 

registrator@etikprovning.se

+46 (0)10-475 08 00

Thank you for this – let us know if anything is still unclear. Please see our attempt of revising the data-availability statement accordingly.

4) We note that you have included the phrase “data not shown” in your manuscript. Unfortunately, this does not meet our data sharing requirements. PLOS does not permit references to inaccessible data. We require that authors provide all relevant data within the paper, Supporting Information files, or in an acceptable, public repository. Please add a citation to support this phrase or upload the data that corresponds with these findings to a stable repository (such as Figshare or Dryad) and provide and URLs, DOIs, or accession numbers that may be used to access these data. Or, if the data are not a core part of the research being presented in your study, we ask that you remove the phrase that refers to these data.

This is not an essential part of our study and we removed this sentence, page 7, last paragraph, line 153-154.. 

Reviewers' comments:

Reviewer's Responses to Questions

Comments to the Author

1. Is the manuscript technically sound, and do the data support the conclusions?

Reviewer #1: Yes

Reviewer #2: Yes

2. Has the statistical analysis been performed appropriately and rigorously? 

Reviewer #1: Yes

Reviewer #2: Yes

3. Have the authors made all data underlying the findings in their manuscript fully available?

Reviewer #1: Yes

Reviewer #2: Yes

4. Is the manuscript presented in an intelligible fashion and written in standard English?

Reviewer #1: Yes

Reviewer #2: Yes

5. Review Comments to the Author

Reviewer #1: The study emphasizes the importance of integrating a palliative care approach, including conversations about care goals and preferred place of death. In addition, better psychosocial support and care coordination are important factors in avoiding acute care at EOL. Another interesting change was the fact that a higher percentage of patients died in non-acute settings. In terms of the organization of EOL care, the results show an increase in the use of SPC at home over time. Despite this, our findings raise questions about how well-integrated care is, as the use of SACTs, ED visits, ICU care and death in other care settings has increased. the study found that communication between acute and palliative health organizations is inadequate and has an impact on unplanned acute hospital admissions.

Research which I hope will be a stimulus for health care organizations and stakeholders to better organize and coordinate care for vulnerable patients

Thank you for this, and we agree. 

Reviewer #2: An extremely clear and well written report.

I do wonder about the availability of data regarding provision of care and potential differences between 2010 and 2017: E.g. in the discussion section, page 11 starting in line 244: could authors comment regarding provision of care, mainly a) healthcare professionals palliative care training differences between 2010 and 2017 and b) the existing number of home care teams between 2010 and 2017 across the region and if these could have affected results in any way

Thank you for this valuable comment; we have tried to elaborate on this, page 13, first paragraph now in lines 273- 278, in the discussion section. 

There are quite a few typos. This is by no means an exhaustive list.

Page 3 line 53 “and to be to able prepare and discuss the purpose and priorities at EOL”

Page 3 line 57 “but a set of measures is widely used”

Page 4 line 86 “Diagnostic data were retrieved”

Page 4 line 92 “patients with lung or pancreatic cancer who died in Stockholm”

And so on

Thank you for this, the manuscript has been carefully revised throughout for typos. 

6. PLOS authors have the option to publish the peer review history of their article (what does this mean?). If published, this will include your full peer review and any attached files.

We agree to publish peer review history. 

Do you want your identity to be public for this peer review? For information about this choice, including consent withdrawal, please see our Privacy Policy.

Reviewer #1: Yes: Cascioli Marta

Reviewer #2: No

---

## [Decision Letter · Decision Letter 1]

1 Jul 2021

Health care utilization at end of life among patients with lung or pancreatic cancer. Comparison between two Swedish cohorts

PONE-D-21-08049R1

Dear Dr. Ullgren,

We’re pleased to inform you that your manuscript has been judged scientifically suitable for publication and will be formally accepted for publication once it meets all outstanding technical requirements.

Kind regards,

Amir Radfar, MD,MPH,MSc,DHSc

Academic Editor

PLOS ONE

Additional Editor Comments (optional):

Reviewers' comments:

Reviewer's Responses to Questions

**Comments to the Author**

1. If the authors have adequately addressed your comments raised in a previous round of review and you feel that this manuscript is now acceptable for publication, you may indicate that here to bypass the “Comments to the Author” section, enter your conflict of interest statement in the “Confidential to Editor” section, and submit your "Accept" recommendation.

Reviewer #1: All comments have been addressed

Reviewer #2: All comments have been addressed

2. Is the manuscript technically sound, and do the data support the conclusions?

Reviewer #1: Yes

Reviewer #2: Yes

3. Has the statistical analysis been performed appropriately and rigorously? 

Reviewer #1: Yes

Reviewer #2: Yes

4. Have the authors made all data underlying the findings in their manuscript fully available?

Reviewer #1: Yes

Reviewer #2: Yes

5. Is the manuscript presented in an intelligible fashion and written in standard English?

Reviewer #1: Yes

Reviewer #2: Yes

6. Review Comments to the Author

Reviewer #1: I find the article interesting and above all I hope it is a stimulus to improve the implementation of palliative care networks and increase communication between the hospital and the local area.

Unfortunately, there is still a large gap both for cancer patients (as demonstrated by the study) and for non-cancer patients

Reviewer #2: All points by the editors and reviewers were addressed. The paper is improved. I have no further comments.

7. PLOS authors have the option to publish the peer review history of their article (what does this mean?). If published, this will include your full peer review and any attached files.

Reviewer #1: **Yes: **Cascioli Marta

Reviewer #2: No

---

## [Editor Report · Acceptance letter]

6 Jul 2021

PONE-D-21-08049R1 

Health care utilization at end of life among patients with lung or pancreatic cancer. Comparison between two Swedish cohorts 

Dear Dr. Ullgren:

I'm pleased to inform you that your manuscript has been deemed suitable for publication in PLOS ONE. Congratulations! Your manuscript is now with our production department. 

Kind regards, 

on behalf of

Dr. Amir Radfar 

Academic Editor

PLOS ONE